# Developing a UK registry to investigate the role of cardiovascular magnetic resonance (CMR) in patients who activate the primary percutaneous coronary intervention (PPCI) pathway: a multicentre, feasibility study linking routinely collected electronic patient data

Rachel C Brierley,[1] Maria Pufulete,[1] Jessica Harris,[1] Chiara Bucciarelli-Ducci,[2] John P Greenwood,[3] Stephen Dorman,[2] Richard Anderson,[4] Chris A Rogers,[1] Barnaby C Reeves[1]

For numbered affiliations see end of article.

**Correspondence to**
Dr Maria Pufulete;
maria.pufulete@bristol.ac.uk

## ABSTRACT

**Objectives** To determine whether it is feasible to set up a national registry, linking routinely collected data from hospital information systems (HIS), to investigate the role of cardiovascular magnetic resonance (CMR) in patients who activate the primary percutaneous coronary intervention (PPCI) pathway.

**Design** Feasibility prospective cohort study, to establish whether: (1) consent can be implemented; (2) data linkage and extraction from multiple HIS can be achieved for >90% of consented patients; (3) local data can be successfully linked with hospital episode data (Hospital Episode Statistics, HES; Patient Episode Database for Wales, PEDW) for >90% of consented patients and (4) the proportion of patients activating the PPCI pathway who get a CMR scan is ≥10% in hospitals with dedicated CMR facilities.

**Participants** Patients from four 24/7 PPCI hospitals in England and Wales (two with and two without a dedicated CMR facility) who activated the PPCI pathway and underwent an emergency coronary angiogram.

**Results** Consent was successfully implemented at all hospitals (consent rates ranged from 59% to 74%) and 1670 participants were recruited. Data submission was variable: all hospitals submitted clinical data (for ≥82% of patients); only three hospitals submitted biochemistry data (for ≥98% of patients) and echocardiography data (for 34%–87% of patients); only one hospital submitted medications data (for 97% of patients). At the two CMR centres, 14% and 20% of patients received a CMR scan. Data submitted by hospitals were linked with HES and PEDW for 99% of all consented patients.

**Conclusion** We successfully consented patients but obtaining individual, opt-in consent would not be feasible for a national registry. Linkage of data from HIS with

**Strengths and limitations of this study**

► We established a successful collaboration among four hospitals and exceeded the recruitment target for the study.
► We identified the main challenges that influence the feasibility of setting up the registry: (1) we could not identify all the eligible population; (2) we did not identify an optimal method of obtaining consent and (3) data from multiple hospital information system were not universally available.
► We did not have the time and resource to implement strategies to overcome these challenges, for example, testing out novel methods of identifying the eligible population and obtaining consent or obtaining the data.
► We did not include patients presenting with a broader diagnosis of acute coronary syndrome but not requiring emergency coronary angiography, who may also benefit from cardiovascular magnetic resonance.

hospital episode data was feasible. However, data from HIS are not uniformly available/exportable and, in centres with a dedicated CMR facility, some referrals for CMR were for research rather than clinical purposes.

## INTRODUCTION

The amount of data about 'usual care' provided by UK National Health Service (NHS) hospitals that is being collected and stored electronically in hospital information

systems (HIS) is rapidly increasing. Previous studies have shown that it is possible to use multiple HIS to retrieve and compile these patient-level data into a research database.[1–3] We conducted a study to test the feasibility of setting up a large multicentre registry for patients who activate the primary percutaneous coronary intervention (PPCI) pathway (patients who have had an ST-elevation myocardial infarction (STEMI) and require an emergency angiogram) by linking data from multiple HIS within several hospitals. If the registry was feasible, we had the longer-term aim to use registry data to evaluate the effectiveness and cost-effectiveness of cardiovascular magnetic resonance (CMR) imaging in this population. There are over 60 PCI centres in the UK offering PPCI (24/7 service), which would be eligible for inclusion in the registry.[4]

CMR is a non-invasive imaging technique that assesses heart structure and function with high spatial and temporal resolution.[5] The use of CMR has increased in recent years in all subgroups of patients with acute coronary syndrome (ACS), including those who activate the PPCI pathway.[6] However, there are no studies comparing health outcomes for patients who did or did not have CMR in patients who do activate the PPCI pathway. Recent systematic reviews highlighted the lack of high quality, adequately powered studies to establish the prognostic value of CMR findings in these patients.[7 8] Similarly, few studies have assessed how CMR changes patient management, despite the fact that cardiologists believe that CMR brings about important changes in management.[9] A well-designed registry would provide information on CMR use, CMR uptake over time, patients' characteristics associated with being referred for CMR, the impact of CMR on patient management and the prognostic value of CMR.

Our feasibility study had several objectives: (1) to implement consent and establish the likely patient consent rate for a future registry; (2) to determine whether data linkage and extraction from HIS could be carried out across multiple PPCI Hospitals in the UK; (3) to determine whether HIS data could be linked to hospital episode data (Hospital Episode Statistics, HES; Patient Episode Database for Wales, PEDW) and (4) to estimate the proportion of patients activating the PPCI pathway who get a CMR scan.

## METHODS
### Study population and eligibility criteria
We approached all patients with ACS who activated the PPCI pathway at four NHS hospitals in England and Wales hosting 24/7 PPCI centres—the Bristol Heart Institute (University Hospitals Bristol NHS Foundation Trust), Leeds General Infirmary (Leeds Teaching Hospitals NHS Trust), Morriston Hospital (Abertawe Bro Morgannwg University Health Board, Swansea) and University Hospital of Wales (Cardiff and Vale University Local Health Board). Two of these (Bristol and Leeds) were

defined as 'CMR centres', that is, hospitals that had a dedicated CMR service; the other two (Swansea and Cardiff) were defined as 'non-CMR centres', that is, hospitals that did not have access to a dedicated CMR service. Patients were included if they were 18 years of age or older and underwent an emergency angiogram, defined as within 2 hours of arrival at the hospital unless specified otherwise by local protocols. Patients were excluded if they were prisoners or lacked mental capacity to consent.

### Study procedures
Eligible patients were identified manually (eg, by checking catheter laboratory records and handover sheets or by checking the local cardiology database or both). Patients were approached with information about the study once they had recovered sufficiently from their angiogram, before they were discharged from hospital. If this was not possible, patients were sent study information and consent forms by post. Postal consents were managed locally at each hospital. We requested participants' consent to use all the information collected by the hospital in relation to the patients' suspected myocardial infarction (index procedure) and all the information related to subsequent care and vital status in the first 12 months following the index procedure (from HES and PEDW).

### Data collection
Each hospital was asked to provide the following data from local HIS for all recruited patients: (a) basic demography, (b) clinical characteristics on presentation at the index admission, periprocedural and postprocedural (PPCI) characteristics, (c) echocardiography and CMR reports, (d) biochemistry and (e) medications on discharge. Each hospital had the option of submitting a linked dataset (ie, link data from different HIS within a hospital) or submit unlinked data extracts from each of the above categories with a unique patient identifier in each extract. We initially established a data linkage model at Bristol (by liaising with the appropriate HIS managers at University Hospitals Bristol NHS Foundation Trust) and shared this with the other hospitals, although the decision about how to submit the data was left to each participating hospital.

For subsequent inpatient and outpatient activity in the 12 months following the index PPCI admission, we applied to NHS Digital to link our dataset with HES (inpatient, outpatient, accident and emergency and critical care data) for the hospitals in England and to NHS Wales Information Service (NWIS) for PEDW data (which collects equivalent information) for the hospitals in Wales. NHS Digital routinely links HES with Office for National Statistics (ONS) mortality data. Mortality data were unavailable through NWIS and the two Welsh hospitals provided these data for their patients. Follow-up data were used to characterise the frequency and duration of outpatient follow-up, unscheduled cardiac-related hospital readmissions and cardiac-related investigations/procedures such as diagnostic angiography, repeat PCI or cardiac surgery.

For patients who declined to consent or did not respond to an invitation to consent before study recruitment ended, we collected a limited anonymised dataset to characterise the non-consenting patient population. Hospitals were provided with a look-up table for either English or Welsh postal codes so that the anonymised datasets could contain information on the Index of Multiple Deprivation (IMD) and Welsh Index of Multiple Deprivation (WIMD),[10] which combines information from several domains (eg, income, employment, education, health, etc) to produce an overall relative measure of deprivation.

## Main outcome measures

The main prespecified outcome measures were feasibility parameters (criteria that would have to be met before making the decision to progress to setting up a full registry): (1) patient consent implemented at all four hospitals; (2) data linkage and extraction from multiple local HIS achieved for >90% of consented patients at all four hospitals; (3) local data successfully linked with HES and PEDW for >90% of consented patients at all four hospitals and (4) CMR scan requested and carried out for ≥10% of patients activating the PPCI pathway in CMR hospitals.

## Data analysis

Data were analysed using Stata/IC V.13. The analysis of feasibility outcomes was descriptive, with quantitative results expressed as percentages and 95% CIs and feasibility assessed by comparing descriptive statistics against the corresponding criteria. We compared patients who did not consent (and for whom we received anonymised data) with those who did consent using standardised mean differences (SMDs).[11] We used means and SDs, and medians and IQRs, for continuous variables and number (per cent) for categorical variables, to report the demographic variables and baseline characteristics of patients recruited from the four hospitals.

## RESULTS

### Implementing patient consent

The flow chart of patient recruitment and data collection is shown in figure 1. Consent was successfully implemented at all hospitals between May 2013 and September 2014, with consent rates ranging from 59% to 74%. Anonymised data for patients identified as eligible but who did not consent were available for over 80% of patients from three hospitals but only 50% of patients from one hospital. Table 1 shows the baseline characteristics of consented and non-consented patients. The IMD and WIMD were slightly higher among those who consented compared with those who did not (SMDs 0.31 and 0.27, respectively). The percentage of patients who received PPCI was also higher among those who consented compared with those who did not (90% vs 65%, respectively, SMD, 0.70).

Table 2 shows the characteristics of participants by hospital. Participants from the four hospitals were similar with respect to demographics, comorbidities and previous cardiac interventions, although rates of hypertension were lower in patients from hospital B compared with patients from hospitals A, C and D (18% vs 41%–52%).

### Data linkage and extraction across hospitals

Figure 2 shows the flowchart of data submitted from each hospital. One of the hospitals provided the data in a linked format; linkage was undertaken centrally for the other three hospitals with data linked on local hospital identifier. Clinical data were available for 93% of patients at all hospitals (range 82%–98%) (figure 2). The availability of echocardiography data was variable; one hospital was unable to submit any echocardiography data, one hospital submitted data for only 34% of patients and two submitted data for over 75% of patients. Echocardiography data were submitted as 'free-text' reports, rather than coded data. Biochemistry data were available for ≥98% of patients from three hospitals; one hospital was unable to submit biochemistry data for any of its patients. Data about medications on discharge were available for 97% of patients from one hospital; the other three hospitals were unable to submit these data.

### Data linkage with HES and PEDW

Identifiers and evidence of signed consent were submitted via secure portal to NHS Digital and NWIS; HES, ONS and PEDW data were received with our unique patient identifier embedded within each file for linkage. Data linkage with HES, PEDW and vital status was achieved for 99% of patients across all hospitals (n=1655/1670). The index PPCI was successfully identified for 1612/1670 (97%) of the patients with linked data, with the admission date in HES/PEDW matching index event date (±1 day). Table 3 shows the cardiac-related events in the 12 months following the index admission in the entire cohort. Overall, 55/1638 (3%) of patients died during follow-up (median age at death 73 years, IQR 66–82). Major adverse cardiovascular events rate in this cohort was 13% (219/1638).

### Proportion of eligible population receiving CMR scan

A CMR scan was performed (within 10 weeks of the index event) for 20% and 14% of consented patients in CMR centres. CMR data from both hospitals were submitted as free-text reports; individual CMR parameters were difficult to extract from these. The baseline characteristics of patients who did and did not receive CMR are shown in table 4. Patients referred for CMR were slightly younger that those who did not have CMR. No other differences were noted.

## DISCUSSION

We tested the feasibility of setting up a large multicentre registry using routinely collected data from HIS to

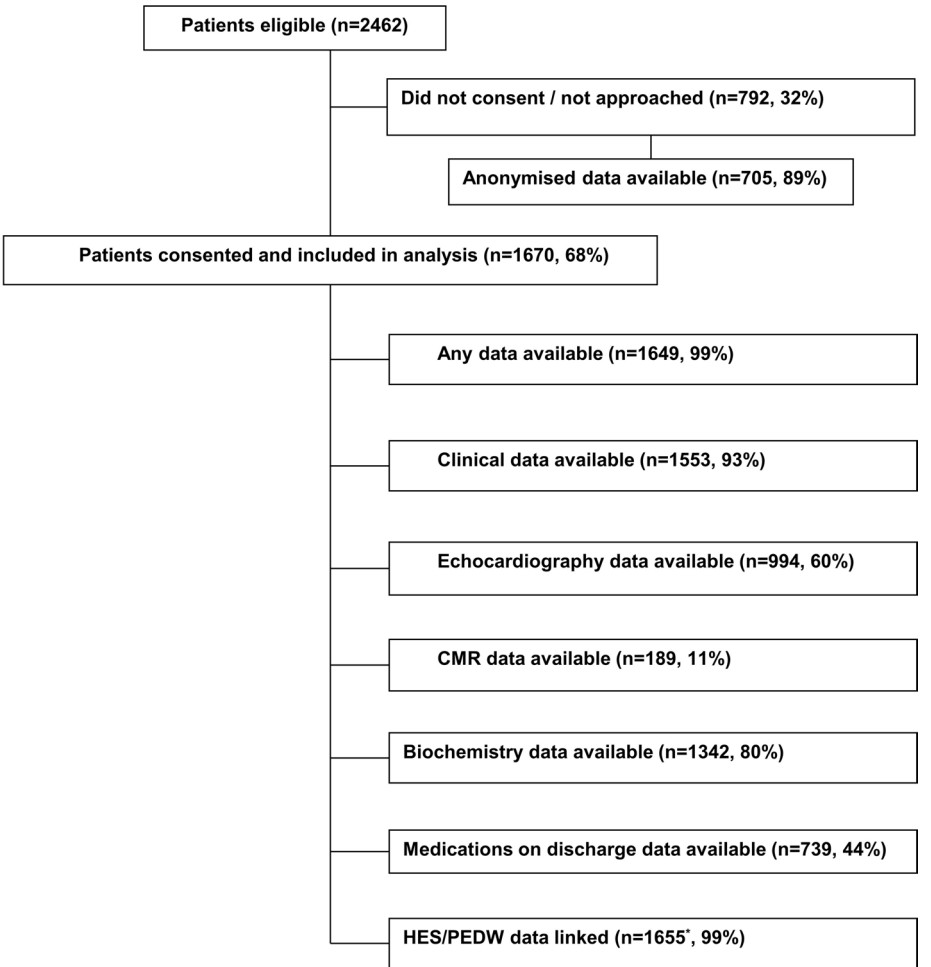

**Figure 1** Flow chart of patient recruitment and data collection. *Number of patients with their index procedure identified in HES/PEDW (exact match on day or 1 day out) or inpatient or outpatient or accident and emergency data available in the year following the index procedure. Note that 1612/1670 (97%) of patient had their index procedure identified in HES/PEDW (exact match on day or 1 day out). CMR, cardiovascular magnetic resonance; HES/PEDW, Hospital Episode Statistics/Patient Episode Database for Wales.

evaluate the effectiveness and cost-effectiveness of CMR in patients with ACS who activate the PPCI pathway. Such a registry would document CMR activity in these patients and provide information on CMR use, uptake over time, patients' characteristics associated with being referred for CMR, the impact of CMR on patient management and the prognostic value of CMR after PPCI. We found that recruitment and consenting of our patient population (using conventional consent), and linkage of local data with hospital episode/vital status data are feasible. More than 10% of patients (20% and 14% in hospitals A and B, respectively) had CMR within 10 weeks of the index event in hospitals with a dedicated CMR facility. However, we did not identify all eligible patients and could not obtain all of the requested data from all hospitals.

### Identifying the eligible population

Patients who activated the PPCI pathway and received PCI were easy to identify from catheter laboratory databases. Nevertheless, there were discrepancies between the number of PPCI patients identified as eligible and approached at each hospital and the numbers of PPCI

cases reported to the British Cardiovascular Intervention Society National Audit of PCIs[4] for each hospital over the recruiting period, indicating that not all eligible patients were identified and approached. Patients who activated the PPCI pathway but had an emergency angiogram without PCI (ie, those with unobstructed coronary arteries) were more difficult to identify. There was no consistent terminology used to describe these patients in the databases; this created confusion and further manual checks, so many patients were missed.

Each hospital implemented its own methods for identifying and consenting patients. Most eligible patients were identified manually (eg, by checking catheter laboratory records or by checking the local cardiology database, or both). One of the participating hospitals set up a database query (including the search terms 'PPCI', 'QueryNot' and 'QuerProc') in the local cardiology database to flag up eligible patients. However, the query was broad and relied on data input by staff in the catheter laboratory, which changed in 'real time'. This made identification of some eligible patients difficult. For example, if a patient

**Table 1** Baseline characteristics of consenting and non-consenting patients (for whom data were available)

| | Consenting patients (n=1649) | Non-consenting patients (n=705) | SMD* |
|---|---|---|---|
| Age (mean, SD)† | 64 (12.7) | 65.0 (14.2) | 0.06 |
| Male sex | 1159/1516 (76%) | 481/677 (71%) | 0.12 |
| Current smoker) | 478/1473 (32%) | 240/602 (40%) | 0.16 |
| IMD score (median; IQR)‡ | 13.4 (7.7–25.8) | 19.9 (10.4–34.9) | 0.31 |
| WIMD rank (median; IQR) § | 891 (459–1370) | 671 (268–1221) | 0.27 |
| Diabetes | 231/1536 (15%) | 134/645 (21%) | 0.15 |
| Hypertension | 601/1455 (41%) | 263/663 (40%) | 0.03 |
| Previous PCI | 157/1552 (10%) | 73/652 (11%) | 0.04 |
| Previous CABG | 37/1540 (2%) | 25/654 (4%) | 0.09 |
| Previous MI | 193/1547 (12%) | 101/650 (16%) | 0.09 |
| PPCI | 1307/1452 (90%) | 429/664 (65%) | 0.70 |
| No PPCI | 145/1452 (10%) | 235/664 (35%) | 0.70 |
| Length of stay (median; IQR) days¶ | 3 (2–4) | 2 (1–3.9) | 0.13 |

Values are numbers (percentages) unless otherwise stated.
*SMDs can be interpreted as follows: values of 0.20 are 'small' in magnitude, those around 0.50 are 'medium' and those around or above 0.80 are 'large'.[27]
Missing data (consented patients, non-consented patients): †(8, 5), ‡(179, 15), §(92, 47), ¶(250, 78).
CABG, coronary artery bypass grafting; IMD, Index of Multiple Deprivation (low value reflects low level of multiple deprivation); MI, myocardial infarction; PCI, percutaneous coronary intervention; PPCI, primary percutaneous coronary intervention; SMD, standardised mean difference; WIMD, Welsh Index of Multiple Deprivation (low value reflects high level of multiple deprivation).

was originally marked as 'PPCI' but was subsequently found to have unobstructed coronary arteries on the angiogram, the 'PPCI' label was removed, but no further information was added, necessitating manual searches by local research staff to identify whether the patient was eligible or not. We did not explore the possibility of using

**Table 2** Baseline characteristics of participants by hospital

| | Hospital A (n=762) | Hospital B (n=272) | Hospital C (n=315) | Hospital D (n=300) |
|---|---|---|---|---|
| Age (mean; SD)* | 65.1 (12.7) | 62.9 (12.5) | 63.8 (12.4) | 63.4 (13.0) |
| Male sex | 512/655 (78%) | 183/246 (74%) | 238/315 (76%) | 226/300 (75%) |
| Current smoker | 191/660 (29%) | 95/240 (40%) | 95/300 (32%) | 97/273 (36%) |
| IMD score (median; IQR)† | 12.2 (7.1–21.7) | 20.5 (10.1–39.2) | – | – |
| WIMD rank (median; IQR)‡ | – | – | 817 (334–1478) | 902 (553–1314.5) |
| Diabetes | 91/700 (13%) | 33/243 (14%) | 70/310 (23%) | 37/283 (13%) |
| Hypertension | 275/663 (41%) | 42/238 (18%) | 143/282 (51%) | 141/272 (52%) |
| Previous PCI | 71/701 (10%) | 35/242 (14%) | 27/310 (9%) | 24/299 (8%) |
| Previous CABG | 21/704 (3%) | 7/243 (3%) | 5/308 (2%) | 4/285 (1%) |
| Previous MI | 88/707 (12%) | 40/240 (17%) | 33/312 (11%) | 32/288 (11%) |
| PPCI | 597/655 (91%) | 220/243 (91%) | 267/295 (91%) | 223/259 (86%) |
| No PPCI | 58/655 (9%) | 23/243 (9%) | 28/295 (9%) | 36/259 (14%) |
| Length of stay (median; IQR) days§ | 3 (2–4) | 2 (1–3) | 3 (1–4) | 2 (1–4) |

Values are numbers (percentages) unless otherwise stated.
The total number of participants in each hospital are patients with any data available.
Missing data by hospital (hospital A, hospital B, hospital C, hospital D):
*Eight patients (1, 0, 0, 7).
†One hundred and seventy-nine patients (169, 10, not collected for hospitals C and D).
‡Data missing for 92 patients (not collected for hospitals A and B, 80, 12).
§Data missing for 250 patients (109, 2, 17, 122).
CABG, coronary artery bypass grafting; IMD, Index of Multiple Deprivation (low value reflects low level of multiple deprivation); MI, myocardial infarction; PCI, percutaneous coronary intervention; PPCI, primary percutaneous coronary intervention; SMD, standardised mean difference; WIMD, Welsh Index of Multiple Deprivation (low value reflects high level of multiple deprivation).

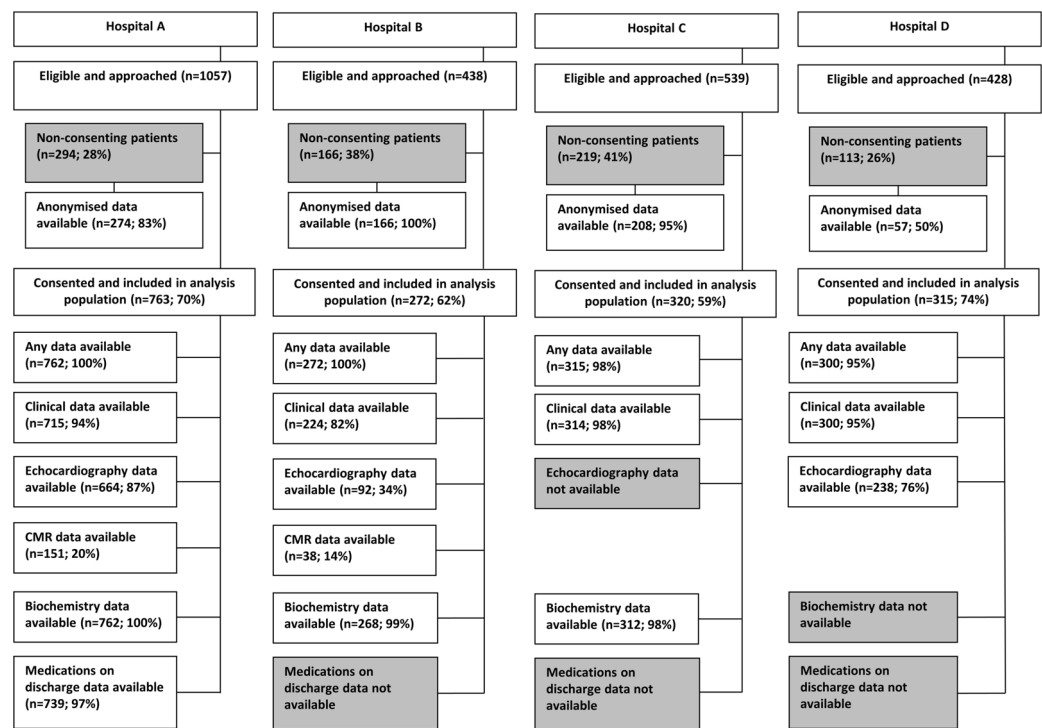

**Figure 2** Flow chart of data availability. Percentages for consenting and non-consenting patients are for those eligible and approached; anonymised data percentages are for total number of patients who did not consent; data available are for those consented. CMR, cardiovascular magnetic resonance.

automated tools such as Natural Language Processing to identify subgroups of our eligible population (eg, those with unobstructed coronary arteries at angiography) in local cardiology databases.

### Implementing consent

We asked all participating hospitals to implement a system of written consent so that we could establish a consent rate and ensure that we had consent to obtain follow-up data from HES and PEDW. We met the feasibility objective of implementing conventional prospective consent at all hospitals. Consent rates, ranging from 59% to 74%, were lower than we had anticipated for a study in which participants have no active involvement, suggesting that many patients identified as eligible were missed. PPCI patients are asleep immediately following the procedure, rarely confined to bed after the first 4–6 hours and many are discharged within 24 hours or transferred to referring hospitals. Some were admitted at the weekend, when research nurse cover was variable. We tried to capture missed patients through postal consent, but return rates were low and we did not have the resources to send reminders or make telephone calls. Competing studies were cited as an issue in some hospitals.

Written consent is resource intensive and, as our data illustrate, would not be feasible for identifying consecutive eligible patients for an extended registry. We explored alternative methods of obtaining consent as part of the feasibility study. We considered presumed consent but concluded that it would be unacceptable legally, given that in principle it is possible for patients to be asked to give consent for their data to be used for the study[12]; exemptions from this rule

are only granted on a temporary basis or when consent is impossible, not just infeasible. We considered the option of requesting consent at the same time as requesting procedural consent (eg, by adding a paragraph to the standard NHS form), but the emergency nature of emergency angiography means that often only verbal assent is recorded by hand in the patient notes.

A possible solution would be to implement changes in NHS information technology (IT) administrative systems (ideally at the national level, maintained locally by NHS or research staff) to allow a record of verbal patient consent to be recorded within their electronic notes (eg, a 'consent module'). The use of a system embedded within the NHS administrative systems could be applied more broadly to all registry projects, so that patients could be easily identified as having given/not given consent, having withdrawn consent or not having been asked for consent. It could also capture whether the patient consents to all data being used and if not, which data are specifically excluded from the consent. However, we are not clear whether verbal patient consent recorded in this way would comply with NHS Digital's requirement for 'explicit' patient consent (the legal basis for accessing HES/ONS data).[13]

Patients and the public are broadly supportive of the use of electronic patient data for research,[14–17] but there are concerns about potential breaches of privacy and misuse of health data.[18] The public response to care data[19] highlighted important challenges in terms of patient confidence and trust in how electronic health records are used in medical research.[20] Newer consent models have been

**Table 3** Rates of death and cardiac-related events in the 12 months following the index admission in the entire cohort

|  | No of events (%) | Person-years of observation | Events per 1000 person-years (95% CI) |
|---|---|---|---|
| Death | 55/1638 (3) | 1536.33 | 35.8 (27.0 to 46.6) |
| MI | 74/1638 (5) | 1376.04 | 53.8 (42.2 to 67.5) |
| Stroke | 2/1638 (0.1) | 1427.12 | 1.4 (0.02 to 5.1) |
| Repeat PCI | 117/1638 (7) | 1347.76 | 86.8 (71.8 to 104.0) |
| CABG | 20/1638 (1) | 1417.25 | 14.1 (8.6 to 21.8) |
| MACE (death, MI, revascularisation) | 219/1638 (13) | 1311.44 | 167.0 (145.6 to 190.6) |

The denominator is 1638 since date of index admission was missing for 22 patients and Hospital Episode Statistics/Patient Episode Database for Wales data were unavailable for 10 patients.
CABG, coronary artery bypass grafting; MACE, major adverse cardiovascular events; MI, myocardial infarction; PCI, percutaneous coronary intervention.

proposed, including a dynamic consent model, which would allow patients to control consent electronically through time and receive information about the uses of their data and up-to-date lay summaries of research projects that have used their data.[21] Nevertheless, such a model would require investment in and maintenance of an e-infrastructure, which would be difficult in the current economic climate of the NHS. Others have called for more adaptive governance models depending on the particular circumstances of the proposed research.[22] It was beyond the scope of this study to explore these possibilities for obtaining consent.

### Data linkage and extraction across hospitals
We did not meet the feasibility criterion for data linkage and extraction (achieved for >80% of patients at all four hospitals) from HIS for all the requested datasets. Only one hospital provided all of the requested datasets for >80% of eligible and consented patients. Some hospitals had entire datasets missing; medications on discharge were not available from hospitals B, C and D; information about echocardiography tests were not available from hospital C and biochemistry data were not available from hospital D. There were several reasons for missing data. Some hospitals were unable to extract data from local stand-alone databases

**Table 4** Baseline and CMR characteristics of patients who did and did not have CMR in hospitals A and B

|  | Patients who had CMR (n=189) | Patients who did not have CMR (n=846) | SMD* |
|---|---|---|---|
| Age (mean, SD)† | 61.8 (11.7) | 65.1 (12.8) | 0.26 |
| Male sex | 121/154 (79%) | 574/747 (77%) | 0.04 |
| Current smoker | 52/168 (31%) | 234/732 (32%) | 0.02 |
| IMD score (median; IQR)‡ | 12.4 (7.5–23.5) | 13.5 (7.8–26.3) | 0.08 |
| Diabetes | 19/176 (11%) | 105/767 (14%) | 0.09 |
| Hypertension | 63/166 (38%) | 254/735 (35%) | 0.07 |
| Previous PCI | 18/176 (10%) | 88/767 (11%) | 0.04 |
| Previous CABG | 3/176 (2%) | 25/771 (3%) | 0.09 |
| Previous MI | 20/174 (11%) | 108/773 (14%) | 0.07 |
| PPCI | 140/154 (91%) | 677/744 (91%) | 0.003 |
| No PPCI | 14/154 (9%) | 67/744 (9%) | 0.003 |
| Length of stay (median; IQR) days§ | 2 (2–3) | 3 (2–3) | 0.11 |

Values are numbers (percentages) unless otherwise stated.
*SMDs can be interpreted as follows: values of 0.20 are 'small' in magnitude, those around 0.50 are 'medium' and those around or above 0.80 are 'large'.[27]
Missing data (patients who had CMR, patients who did not have CMR):
†Two patients (1, 1).
‡One hundred and eighty patients (43, 137).
§One hundred and twelve patients (32, 80).
CABG, coronary artery bypass grafting; CMR, cardiovascular magnetic resonance; IMD, Index of Multiple Deprivation (low value reflects low level of multiple deprivation); MI, myocardial infarction; PCI, percutaneous coronary intervention; PPCI, primary percutaneous coronary intervention; SMD, standardised mean difference.

which were designed for a specific purpose, for example, logging biochemistry requests, storing and presenting test results. Such local databases were typically old and did not have an interface for data extraction; data would have had to be retrieved manually for each patient in turn, rather than submitting a 'batch' query. Obtaining data in this way was considered infeasible for a national registry (as well as time consuming) and hospitals were not requested to do so. Some data (eg, medications on discharge, bedside echo-cardiography) were not recorded electronically and were only available in paper form (sometimes scanned but not coded/searchable). Non-bedside imaging tests were available electronically but as free-text reports rather than coded data and individual variables of interest were impossible to extract from these reports.

We had initially envisaged establishing a data linkage and extraction model in one hospital and using this to guide data extraction at the other hospitals. We envisaged that this model would be automated, with minimal IT support beyond the set-up phase. However, this was not possible because HIS for each dataset differed across hospitals and different hospitals provided different levels of start-up IT support for the research project. Data linkage requires expertise in several areas, including knowledge of the databases to be linked and skills in creating linkage programmes. This expertise was variable across hospitals and in-depth knowledge sometimes lacking; not surprising given the diversity of databases involved and the fact that some were old and being phased out.

The complexity of data retrieval from multiple HIS in secondary care in the NHS has been highlighted previously.[1] The government's vision for the NHS, outlined in the National Information Board's complex Personalised Health and Care 2020 framework,[23] is that it will be an entirely digital organisation by 2020. Although such a system, if properly implemented, should provide benefits for research as well as patient care, it is an ambitious goal given the current financial and operational pressures within NHS organisations. The earlier (and equally ambitious) National Programme for IT, which aimed to create a single electronic care record for patients, connect primary and secondary care IT systems, and provide a single IT platform for health professionals was dismantled, largely because IT companies could not provide solutions to match the ambition.[24] Our experience from this study has highlighted the poor quality electronic infrastructure in hospitals and multitude of clinical databases that are not amenable to 'interoperability' (ie, not needing extensive customisation or permission from the database vendor to provide data flow). Full interoperability for HIS therefore still appears a distant goal.[25]

## CMR rates in hospitals with dedicated CMR facilities

Although more than 10% of patients in hospitals with a dedicated CMR facility were referred for CMR, many of these referrals (33% on hospital A) were for research rather than clinical purposes. It is not clear whether this reflects practice in other hospitals with dedicated CMR facilities. Referral for non-clinical reasons (unless the findings are used to change clinical management) would dilute any observed effect of CMR on outcomes in an extended registry.

## Data linkage with HES and PEDW

We met the feasibility objective with respect to linkage with HES and PEDW. Linkage was achieved for 99% of patients. We failed to achieve linkage for the remainder mainly because of missing identifiers (NHS numbers) in the datasets submitted from the Welsh Hospitals.

## Strengths and limitations

We established a successful collaboration among four hospitals and exceeded the recruitment target for the study of 1600 patients. We also identified the main challenges to establishing a prospective registry in the way we had envisaged: identifying the eligible population is difficult to do from HIS; the conventional consent model fails to capture a sizeable proportion of the eligible population and would be costly to implement; and, most importantly, the required local data from HIS are not uniformly available/exportable.

We are aware that CMR may benefit other groups of patients with ACS (eg, those with non-STEMI). However, we considered at the outset that it would be impossible to identify consecutive patients presenting with a broader diagnosis of ACS but not requiring emergency coronary angiography; being able to define and ascertain the entire eligible population is a key requirement of a high-quality registry. This was a particularly important consideration given that the proposed registry was based on the premise of linking information collected in the course of usual care, avoiding the time and cost burden of primary data collection.

## Conclusions

We conclude that setting up a national registry by linking routinely collected data to investigate the role of CMR in patients with ACS who activate the PPCI pathway is not feasible in the current NHS IT environment. Registries (eg, national audits) that have managed to overcome issues with data collection and quality have achieved this through a lengthy process necessitating many iterations (and a feedback loop) before attaining any quality targets. This requires considerable investment, infrastructure and commitment at the national level, which would be difficult to achieve in a research setting. Although the importance of electronic health records for research has long been recognised and the wealth of information in the NHS puts the UK at a research advantage,[26] this potential has yet to be fully realised. The rapidly changing IT environment in the NHS will likely overcome the challenges of data extraction we experienced. Access to the data is likely to be the main challenge in the future; the lack of clarity over the safe and ethical secondary use of routinely collected data and increased concern about misuse of such data are the biggest barrier preventing better access to routine data for research.

## Author affiliations
[1]Clinical Trials and Evaluation Unit, University of Bristol, Bristol, UK
[2]NIHR Bristol Cardiovascular Research Unit, Bristol Heart Institute, University of Bristol, Bristol, UK
[3]Multidisciplinary Cardiovascular Research Centre and Leeds Institute of Cardiovascular and Metabolic Medicine, University of Leeds, Leeds, UK
[4]Department of Cardiology, University Hospital of Wales, Cardiff, UK

**Acknowledgements** We would like to thank the local research teams who contributed to the study, including Ruth Bowles, Jo Roberts, Alan Davies, Helen Kingsland, Petra Bijsterveld, Kathryn Somers, Richard Gillott, Lesley Davies, Stephen Hiles, Stephen Morris and Clare Fagan.

**Contributors** RCB: conducted the study and compiled a list of the main study challenges. MP: designed and conducted the study and wrote the manuscript. JH: conducted the data linkage for three of the four hospitals, conducted the statistical analysis and drafted some sections of the manuscript. CB-D: conceived the overall feasibility study and provided clinical and CMR expertise. JPG: principal investigator at one of the recruiting hospitals, provided cardiology and CMR expertise. SD, RA: principal investigators at one of the recruiting hospitals, provided cardiology expertise. CAR: provided guidance with respect to the statistical analysis. BCR: chief investigator with overall responsibility for the study, designed the study and provided strategic direction with interpretation of study results. All authors read and approved the final version of the manuscript.

**Funding** This study is funded by the National Institute for Health Research (NIHR) Health Services and Delivery Research (HS&DR 11/2003/58). The British Heart Foundation and NIHR Bristol Biomedical Research Unit for Cardiovascular Disease funded some staff time (MP, JH, CBD, CAR, BCR).

**Disclaimer** The views expressed are those of the authors and not necessarily those of the NHS, the NIHR or the Department of Health.

**Competing interests** JPG has received a research grant from Philips Healthcare.

**Patient consent** Not required.

**Ethics approval** Ethical approval for the study was obtained from National Research Ethics Service (NRES) Committee South West-Central Bristol.

**Provenance and peer review** Not commissioned; externally peer reviewed.

**Data sharing statement** Anonymised individual patient data (excluding hospital episode/vital status linked data) can be made available for secondary research, conditional on assurance from the secondary researcher that the proposed use of the data is compliant with the MRC Policy on data preservation and sharing regarding scientific quality, ethical requirements and value for money. We are prevented from sharing hospital episode data/vital status data under our data sharing agreement with NHS Digital. Please contact maria.pufulete@bristol.ac.uk to discuss any data requests.

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
