## [Reviewer comments · BMJ Open]

ARTICLE DETAILS

TITLE (PROVISIONAL)	Is it feasible to develop a national registry by linking routinely collected electronic patient data to investigate the role of cardiovascular magnetic resonance (CMR) in patients who activate the PPCI pathway?
AUTHORS	Brierley, Rachel Pufulete, Maria Harris, Jessica Bucciarelli Ducci, Chiara Greenwood, John Dorman, Stephen Anderson, Richard rogers, chris Reeves, Barnaby

VERSION 1 - REVIEW

REVIEWER	Navdeep Gupta
REVIEW RETURNED	11-Sep-2017

GENERAL COMMENTS	The authors describe the feasibility of developing a registry using electronic health records(EHRs) to investigate the role of Cardiovascular magnetic resonance in patients activating primary PCI pathway. This a very relevant and timely study as EHRs are increasingly used by health care facilities for managing patient care data. Use of EHRs to conduct research is promising however the potential is not fully utilized with legacy systems with interconnectivity, missing data and consent issues. The authors have highlighted some of the local issues in setting up such a research dataset. A few questions/suggestions regarding the manuscript: • The authors mention that being able to define and ascertain the entire eligible population is a key for registry. Manual identification of patients for inclusion might not be the most efficient for an EHR based registry. Did the authors attempt any automated batch query for inclusion? An example would be using ICD9 procedure codes for selection of patients undergoing coronary angiograms. What were the issues preventing implementation of the same? The authors describe that there was no consistent terminology used to describe patients without PCI in databases, were the description of the procedures text-based and therefore available automated tools could not be employed. It might be feasible to employ Natural Language Processing to improve capture of these patients.• I would suggest briefly describing the Index of Multiple Deprivation or having appropriate references/websites mentioned in bibliography for non-British readers• Please remove bracket on Line 46, page 13
--

REVIEWER	Joao H Bettencourt-Silva
REVIEW RETURNED	26-Oct-2017

GENERAL COMMENTS	This paper describes the efforts in assessing the feasibility of developing a national registry using electronic data from four UK hospitals. This is a well written and compelling article that takes the reader through the challenges of obtaining information on a given clinical domain (ACS and use of CMR) for research and other secondary uses. The authors conclude that, given their set of criteria, it is not at present feasible to develop a national prospective registry, and some recommendations and suggestions are made. The authors defined and tested four main outcome measures to make this assertion: 1) consent implemented, 2) data availability (>90%), 3) linkage with HES (>90%), 4) CMR scans for >=10% patients in PPCI pathway. The most limiting factor according to the authors was 2) the lack of data. Scientifically, the challenges pertaining to informatics infrastructures have long been reported in the literature and perhaps even more so in health/clinical informatics venues. Although it may still be relevant to point out and report on these issues, as we would expect to have seen improvements over the last decade, the main contributions would perhaps be recommendations and suggestions for improvement. In particular, what would need to be in place for this exercise to be carried out successfully? It would be beneficial if this could be made clearer in the manuscript, perhaps in the discussion, as a set of bullet points. In addition, could the authors please expand on the use of 90% as a threshold value? One key point that the authors may consider discussing is that of future work. There have been several national initiatives aiming to collect routine information from hospitals in the UK (e.g. national audits) and what we have learnt from them is that obtaining good quality routine information from hospitals is a lengthy process requiring a number of iterations (and a feedback loop) before achieving any quality targets. In fact, much of the information from these initiatives would not be previously collected or standardised and it is now thanks to them that richer, more accurate information has begun to be available in hospitals and registries. This reviewer wonders if this would also be the case for the registry proposed in this paper and could the authors please explain any limiting factors for continuing this work over a longer period of time. A few other suggestions to consider:  • Could the authors state how many hospitals across the UK would be eligible to be incorporated in such a registry? It is important to gain perspective of the sample of hospitals that the authors selected in this feasibility study in a broader context. Across the UK, some institutions may be more technologically oriented than others (i.e. having integrated EMRs) and this could have an effect on the availability of information. It could also be of interest to investigate the proportion of required information that exists in HIS but can't be extracted and the proportion that is not being collected on a routine basis.
--

	 • Could the authors also briefly comment/expand on the wider benefit of having such a national registry (in the introduction or conclusions)? • In the introduction, could the authors succinctly explain the PPCI pathway activation? • Regarding outcome 4), the CMR scan rates could be discussed in more detail with respect to the potential bias and limitations introduced if most CMR referrals are indeed being made for non-clinical reasons. • Regarding consent, would it be possible to include the number of those consenting and non-consenting who have done so by post or after recovery? This could be added, for example, as a row in the characteristics table or described in the text. • It is unclear what ethical approval was sought and obtained for this feasibility study and this should be briefly mentioned in the methods section. This reviewer believes that initiatives of this kind should be encouraged and that they help increase the quality and availability of the information in hospitals, leading to further research opportunities. This reviewer would also encourage the authors to share this information in other informatics venues.
--	--

VERSION 1 – AUTHOR RESPONSE

Response to Reviewers' comments:

Reviewer 1

The authors mention that being able to define and ascertain the entire eligible population is a key for registry. Manual identification of patients for inclusion might not be the most efficient for an EHR based registry. Did the authors attempt any automated batch query for inclusion? An example would be using ICD9 procedure codes for selection of patients undergoing coronary angiograms. What were the issues preventing implementation of the same? The authors describe that there was no consistent terminology used to describe patients without PCI in databases, were the description of the procedures text-based and therefore available automated tools could not be employed. It might be feasible to employ Natural Language Processing to improve capture of these patients.

We did attempt an automated batch query for inclusion in the catheter laboratory database at one of the participating hospitals to flag up eligible patients. However, the query had to be broad to capture all the eligible population, and in particular a subset (patients undergoing emergency angiography but found to have unobstructed coronary arteries) that was not described consistently in the database; consequently, research staff had to perform further manual checks. The query also relied on data input by staff in the catheter laboratory, which changed in "real-time", so the information required was often not available at the point of patient recruitment. This is described on page 12/13 of the discussion (Identifying the eligible population).

We were not able to use ICD procedure codes in our setting of obtaining prospective patient consent before hospital discharge, because ICD codes are added to discharge summaries, which were not available to us at the time of recruitment.

We did not explore the possibility of using automated tools such as Natural Language Processing to identify eligible patients in local cardiology databases. We have added a statement to this effect in the discussion (page 13, Identifying the eligible population).

I would suggest briefly describing the Index of Multiple Deprivation or having appropriate references/websites mentioned in bibliography for non-British readers

We have provided an explanation and included a reference as suggested.

Please remove bracket on Line 46, page 13
We have removed the bracket as requested.

Reviewer 2

Scientifically, the challenges pertaining to informatics infrastructures have long been reported in the literature and perhaps even more so in health/clinical informatics venues. Although it may still be relevant to point out and report on these issues, as we would expect to have seen improvements over the last decade, the main contributions would perhaps be recommendations and suggestions for improvement. In particular, what would need to be in place for this exercise to be carried out successfully? It would be beneficial if this could be made clearer in the manuscript, perhaps in the discussion, as a set of bullet points. In addition, could the authors please expand on the use of 90% as a threshold value?

We have highlighted in the last paragraph of the discussion section (entitled Data linkage and extraction, page 16) that the main barrier data linkage / extraction across multiple HIS is poor quality electronic infrastructure and lack of “interoperability”, and that the likely solution to this problem is a unified, entirely digital NHS as outlined in the National Information Board’s complex Personalised Health and Care 2020 framework.

Regarding the 90% threshold, we assume the reviewer refers to our threshold of data linkage and extraction from multiple local HIS achieved for >90% of consented patients. There is no established threshold in the literature regarding an acceptable proportion of missing data, but researchers have suggested that >10% missing data is likely to bias the analysis and weaken the generalisability of the study (e.g. Bennett DA. How can I deal with missing data in my study? Aust N Z J Public Health. 2001;25(5):464–469).

One key point that the authors may consider discussing is that of future work. There have been several national initiatives aiming to collect routine information from hospitals in the UK (e.g. national audits) and what we have learnt from them is that obtaining good quality routine information from hospitals is a lengthy process requiring a number of iterations (and a feedback loop) before achieving any quality targets. In fact, much of the information from these initiatives would not be previously collected or standardised and it is now thanks to them that richer, more accurate information has begun to be available in hospitals and registries. This reviewer wonders if this would also be the case for the registry proposed in this paper and could the authors please explain any limiting factors for continuing this work over a longer period of time.

We agree with the Reviewer that data collection for our proposed registry would improve markedly if we had investment, infrastructure, adequate time and commitment at the national level. However, this would not be possible in a research setting. We have added a sentence to this effect in the Conclusions section (page 17).

Could the authors state how many hospitals across the UK would be eligible to be incorporated in such a registry? It is important to gain perspective of the sample of hospitals that the authors selected in this feasibility study in a broader context. Across the UK, some institutions may be more technologically oriented than others (i.e. having integrated EMRs) and this could have an effect on the availability of information. It could also be of interest to investigate the proportion of required information that exists in HIS but can't be extracted and the proportion that is not being collected on a routine basis.

There are over 60 PCI centres in the UK offering primary PCI (24/7), which would be eligible for inclusion in the registry. We have added a statement to this effect in the Introduction (page 1). We have already highlighted in our manuscript the data that could not be extracted electronically (CMR and echocardiography free text reports); see Results section (Data linkage and extraction across hospitals, and Proportion of eligible population receiving CMR scan, page 7) and Discussion (page 15). All the data required for the registry is collected on a routine basis.

Could the authors also briefly comment/expand on the wider benefit of having such a national registry (in the introduction or conclusions)?

We have added a sentence in the Discussion (page 12) stating that such a registry registry would document CMR activity in PPCI patients and provide information on CMR use, uptake over time, patients’ characteristics associated with being referred for CMR, the impact of CMR on patient management and the prognostic value of CMR after PPCI.

In the introduction, could the authors succinctly explain the PPCI pathway activation?
We have added a statement in the introduction as suggested.

Regarding outcome 4), the CMR scan rates could be discussed in more detail with respect to the potential bias and limitations introduced if most CMR referrals are indeed being made for non-clinical reasons.

We have added a sentence in the Discussion (page 16) stating that referral for non-clinical reasons (unless the findings are used to change clinical management) would dilute any observed effect of CMR on outcomes in an extended registry.

Regarding consent, would it be possible to include the number of those consenting and non-consenting who have done so by post or after recovery? This could be added, for example, as a row in the characteristics table or described in the text.

We are unable to present data for patients who consented by post vs. in hospital because these data was not provided by all hospitals.

It is unclear what ethical approval was sought and obtained for this feasibility study and this should be briefly mentioned in the methods section.

See response to Editorial Requirement (final point).

This reviewer believes that initiatives of this kind should be encouraged and that they help increase the quality and availability of the information in hospitals, leading to further research opportunities. This reviewer would also encourage the authors to share this information in other informatics venues. We thank the reviewer for this comment. The findings of this study have already been presented at several national conferences.

Additional changes to the manuscript

We carefully read through the manuscript and corrected the errors highlighted below. This has no impact on the conclusions of the study, since the tables are purely for descriptive purposes and we did no comparative analyses.

Table 1. The denominators included patients for whom there were some missing data; these patients have now been removed.

Table 2. We noticed incorrect numbers in several rows in the table. This arose because of incorrect transcription.

Table 3. The numbers have changed slightly because we fine-tuned the calculation for the person-years at risk to take into account missing data and events occurring in that year.

VERSION 2 – REVIEW

REVIEWER	Navdeep Gupta
REVIEW RETURNED	02-Jan-2018

GENERAL COMMENTS	I appreciate the time and patience of the authors in addressing reviewer concerns and providing details.
--

REVIEWER	Joao H Bettencourt-Silva
REVIEW RETURNED	04-Jan-2018

GENERAL COMMENTS	The authors addressed the points raised by this reviewer and only very minor remarks remain: - Page 7, Paragraph 1: '75 of patients' seems to be a typo. - Figure 2 flowchart values for patients consented and included in analysis population do not all match the denominators in Table 2 (shown at the top of each column). Could the authors please double
---

check the numbers are correct in this table?
--

VERSION 2 – AUTHOR RESPONSE

Reviewer 2

- Page 7, Paragraph 1: '75 of patients' seems to be a typo.
The typo identified should read "75% of patients".

- Figure 2 flowchart values for patients consented and included in analysis population do not all match the denominators in Table 2 (shown at the top of each column). Could the authors please double check the numbers are correct in this table?

The numbers are correct, reflecting the total number of patients with "Any data available" in Figure 2. We have added a footnote in the table to clarify this.